# Digital Citizenship and Digital Literacy in the Conditions of Social Crisis

**Valentina Milenkova and Vladislava Lendzhova *** 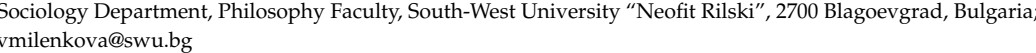

Sociology Department, Philosophy Faculty, South-West University "Neofit Rilski", 2700 Blagoevgrad, Bulgaria; vmilenkova@swu.bg
\* Correspondence: vlendzhova@swu.bg

**Abstract:** In the present day, Internet technology and social media totally dominate as a means of communication. The media and social interaction have a two-sided nature, as the important role is not only those of the media messages to the users, but of the users to the media, too. This article aims to present the dominant importance of digital media, digital literacy transformation into a precondition for social inclusion, and an indicator of professional competence and social skills. Digital citizenship is a term that reflects the level of training and competencies, with a view to active participation in social, professional, and civic life. The article is based on two methods: Focus groups that were conducted in late 2019, which includes: Students, young mothers, pensioners, and unemployed. The second method used is the documents analysis—publications, materials, and quantitative results of research on social reactions to digital media as a source of information in the context of the COVID-19 pandemic crisis. The combination of materials and data that have been analyzed are related to the period of the lockdown between March 2020 and December 2020. In the time of global social crises and confrontations, digital media literacy has turned out to be of critical importance for the normal course of social events and their interpretations. In this regard, digital citizenship contributes to social understanding and control, as well as the individual practices in the global pandemic trajectory.

**Keywords:** digital citizenship; digital literacy; COVID-19; media; trust

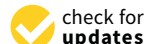



## 1. Introduction

The influence of digital technologies over the structure and dynamics of social relationships fosters the study of problems related to the individual digital literacy, the importance of digital skills and information literacy, and the role of digital literacy for personal development and social integration into society. In Europe, some of the leading institutions have created and put into practice strategies and concepts for training in digital literacy and information [1–3]. In Bulgaria, as part of Europe, there have also been developed and implemented relevant practical projects, documents, and guidance aiming information and gaining experience of digital skills of individuals in the knowledge society.

The aim of this article is to present the dominant importance of digital media, the transformation of digital literacy into a precondition for social inclusion and an indicator of professional competence and social skills. On this basis, digital citizenship is a term that reflects the level of training and competence, in order to actively participate in social, professional, and civil life. Digital citizenship refers to awareness, as well as the ability of sifting out the fake news, and to be critical of social life and what is happening in general. Digital citizenship also means activity and to take a position towards specific events in the life trajectory. In the modern world, society is facing various crises which require a specific approach for managing, along with behavioral strategies. The pandemic crisis caused by COVID-19 can be considered as a concrete example in this direction.

The research questions addressed in this article are: To what extent have digital media become an element of people's lives? How do people evaluate digitization and digital skills?

Is digital media perceived as a reliable source of information in COVID's terms? How do educational institutions manage in a lockdown and online environment? In this context, the article shows that people are starting to perceive digital media as a source of diverse and even contradictory information that requires a critical attitude in terms of weeding out fake news; whereby the digital media impose the requirement to improve the digital media literacy of society and to realize the need for a critical view and attitude towards the surrounding world. The innovativeness of the article is expressed in the widespread digitalization of the social world and restructuring the thinking and evaluations of people of different ages and statuses in accordance with the digital environment, as well as the construction of new learning contexts aimed at creating digital citizens. The empirical results used by various studies are in full harmony with the presented theoretical theses and analysis, giving basic arguments for the ongoing changes in modern society.

## 2. Basic Considerations

### 2.1. Digital Citizenship

The concept of "citizenship" with its present meaning came into our lives in the late 19th and early 20th century [4]. Digital citizenship is relevant to the harmonious coexistence and the promotion of "mutually beneficial development of individuals and communities where they live" [5]. The concept refers to awareness, the development of critical attitudes, thinking, and opinions, as well as the strict fulfillment of responsibilities proceed from different social roles. Digital citizenship is relevant to online information, knowledge of digital technology, creation of content in the digital environment, and following the ethical rules and expectations for online behavior.

Being a digital citizen is more important thing in modern world. Therefore, in education there are some key features to make the students digital citizens for looking at the 21st century digital citizenship goals. These key factors are: Student learning and academic performance, student environment and student behavior, and student life outside the school environment [6]. Internet safety education efforts developed quickly in response to public concern about the potential risks that youth face when online, but the direction of youth education in this area is evolving as more is understood about youth behavior and experiences using new technology. Digital citizenship has three basic online dimensions: Higher wages (resulting from digital skills), democratic participation, and better communication opportunities thanks to internet connections [7]. Early use of the term digital citizenship referred to online access (e.g., "increasing the number of youth digital citizens") [8,9], but it has been used more recently to refer to safe and responsible behavior online. One author defined digital citizenship as comprising the concepts of responsibility, rights, safety, and security [10].

Others describe it as involving "appropriate technology usage," and "making safe, responsible, respectful choices online". A media education program has translated digital citizenship education into curricula on the following topics: Internet safety, privacy and security, relationships and communication, cyberbullying, digital footprints, reputation, self-image and identity, information literacy, and creative credit and copyright. However, if digital citizenship is going to become a new educational focus that is marketed to schools, a significant amount of conceptual and evaluation work is needed to ensure that its goals are well-defined, and its outcomes successfully achieved.

Computer technology is a platform for the creation and distribution of content, providing greater speed in creating and processing data. Based on the accumulation of large information resources, they are establishing themselves as new tools that affect the requirements for citizens, affecting their commitment and fair participation. Digital citizenship is becoming an object of various studies and policy analyses [11], which define the relevant qualifications, these are: Digital access, digital commerce, digital communication and cooperation, digital etiquette, digital governance, digital health and well-being, digital law, digital rights and obligations, and digital security and confidentiality. All specified qualifications are extremely important for the competencies of the digital citizen, as their

important premise is digital literacy and skills. The future belongs to digital citizens, so public policies must minimize technological disparities and divisions in society caused by age, race, ethnicity, income, and education in order to achieve fuller digital participation [7]. The role of education in this process is indisputable [12], especially given the growth of Data Sciences, which focuses on decision-making and processing of large data sets that are applied in all areas of life in the digital marketing environment [13].

Technology provided without direction or instruction has the potential to cause issues with others. In the past, there has been an inability to hold individuals accountable for their actions related to inappropriate use of technology, or to set any standards for appropriate use. Now, with this basic set of skills, steps need to be taken to implement these standards in daily practice in schools.

### 2.2. Digital Skills and Digital Literacy

The concept of digital skills and digital literacy is still new and not adequately researched and described. The complexity and difficulty of the definition come from the fact that digital skills are based on fast-changing information technologies that create problems to define the exact criteria that determine the skills and competences needed, requiring ongoing adaptation to changing practices. For this reason, in the academic literature, digital skills are defined as a dynamic conception. Some of the related terms that are known are: Technological skills, 21st century skills, information literacy, digital literacy, digital competencies [14].

In modern societies, digital technologies are becoming a part of and are integrated in nearly all professions. The characteristics of age are the factors determining the level of digital literacy, as well as the ability to quickly learn and susceptibility to technologies. In this regard, the Australian sociologist Mark McCrindle classifies in his prominent book "*ABC of XYZ: Beyond the Words*" several generations that are complexly distinguished, using many factors, including their digital skills:

- The Elders Generation, which includes people born before 1925;
- the Generation of Builders born between 1925 and 1946;
- a generation of Baby Boomers born between 1946 and 1964;
- Generation X, born between 1965 and 1979 and adapting to the digital;
- the W generation or children of transition born between 1980 and 1994;
- Generation Z or the Millennium Children born between 1995 and 2009;
- Generation Alpha children born after 2010.

The formation of digital skills and digital literacy is a process in which the receiver becomes able to find, understand, evaluate, and apply information in various forms to solve personal, professional, community, regional, social, or even global problems. In order to achieve a high level of digital literacy, the condition of developing universal thinking must be fulfilled [15]. In this aspect, digital literacy is the process of adapting to the digital skills that individuals need to acquire through digital technology. In addition, digital literacy can be defined as a type of social practice that requires the ability to read and write through the use of digital technology. Thus, the digital literacy involves the access, use, and analysis of texts, as well as their creation and distribution.

The development of digital technologies requires that individuals should use a growing range of skills to fulfil tasks and solve problems in the digital environment. These skills are often referred to as "digital literacy" by Gilster [16] which is presented as a particular kind of thinking, allowing users to work intuitively in digital environments, to access easily and effectively a wide range of knowledge inculcate in these environments. Digital literacy is usually understood as a set of technical, procedural, cognitive, and emotional-social skills such as file processing and graphic editing. Ferrari [17] identifies digital literacy as a combination of information skills, communication skills, content creation skills, security skills, as well as skills for problem solving. The operationalization of communication skills is technically oriented based on the number of devices used for online communication.

Other researchers, such as Helsper and Eynon, defined four main categories of qualification: Technical, social, critical and creative skills. This classification is based on studies of media literacy, which suggests that skills must be measured beyond the basic technical level and in terms of the ability to work with communication technologies for social needs. A study by Van Deursen, Helsper, and Eynon provides information on the relationship between operational and navigational information skills, which are stronger in older age groups than in younger. According to them, the differences are related to the ways in which people use the internet and the ways in which they learn to use it, which may differ depending on social and age statuses as well as across different professions.

The concept of "digital skills" as a series of several types of skills was introduced by [14,18]. According to Steyaert, basic or most important are the so called "instrumental skills", or as Van Dijk designates them "operational skills", which include the ability to work with software and hardware. According to Steyaert, there is a difference between "structural skills" and "strategic skills", however, Van Dijk differentiates "information skills" from "strategic skills". "Information skills" include search skills, and choices of processing information in computer and network sources. Potentialities for using computer and network resources as tools for specific purposes include "strategic skills". In the academic literature and in public opinion, these skills are very important, and much attention is paid to them. According to many researchers involved in the processing of information, other types of skills necessary for the successful use of the Internet should not be neglected. For example, Van Deursen classifies six types of digital skills: operational; formal; information; communication; content creation; and strategic.

Some of the classifications focus only on the technical skills of using digital technologies. Generally, it is argued that when measuring digital skills, both basic skills, those needed to use the Internet and those needed to understand and use online content, should be considered [14,19,20]. According to Van Deursen and Van Dijk, the measurement should also include communication and social–emotional skills that are needed in using of social media.

### 3. Methodology

The article is based on two main methods: Focus groups and document analysis. Focus groups refer to two periods of time: In 2019, before the COVID lockdown and after the first COVID lockdown. They include representatives of various social communities and discuss the problems of digitalization and the crisis that society is going through during the COVID-19 pandemic. A total of five focus groups were held.

Three of the focus groups were held during the period of May–September 2019 with: Young mothers, pensioners, and Bulgarian Muslims. The aim of the discussions was to debate key topics related to digital literacy, digitalization, and digital skills in the modern knowledge society, as well as the ways they were formed; the policies and practices that affect them. The indicated target groups present the use of digital resources and the possession of individuals' unequal levels of digital skills and digital literacy. As representatives of different ways of thinking, attitudes, and approaches to digitalization and digital literacy, they also reflect the influence of ethnicity, age, and status of individuals. In this sense, each of the focus groups becomes "a bearer" of some community attitude and demonstrates a specific social profile. We will briefly describe them. In the town of Belitsa, a focus group was held with representatives of Bulgarian Muslims, in the town of Blagoevgrad, the focus group was with representatives of young mothers, and in the town of Kostenets, with representatives of pensioners. The focus groups conducted within the framework of the project Digital media literacy in the context of the "knowledge society: status and challenges". The research is funded by the Bulgarian National Science Fund and headed by prof. Valentina Milenkova.

The three towns were chosen because of their specifics: The town of Blagoevgrad is a regional town, where two universities are situated, South-West University "Neofit Rilski" and American University in Bulgaria. The town of Kostenets is a small town in

the Sofia region, characterized by a low employment rate. For this reason, most of the town's population works in the Sofia capital. The town of Belitsa is also an example of a small town, with a distinctive feature of the mixed population in the area—Muslims and Christians. A total of 22 respondents participated in the three focus groups: 8 Bulgarian Muslims, 7 young mothers, and 7 pensioners. Focus group participants differ in social status, age, and education. The focus group respondents were selected through personal contacts or due to their previous participation in other research projects.

In addition, two focus groups were conducted with students from the University of National and World Economy in Sofia in October 2020 after the first lockdown from March–May 2020. The number of students in each focus group was 10 or a total of 20 respondents. These were first-year students, and men and women have been equally distributed. The respondents were in the age range 19–21 years; these are digitally active generations. The groups (participated in the discussions) were representatives of a similar type of thinking, attitudes, and approach to digitization and digital transformation. The focus groups carried out were saturated with a specific social and functional charge of the lockdown, caused by the COVID-19 situation. In this sense, the focus groups become the bearer of a community attitude and demonstrate a specific social profile of thinking.

The second method used in this article is document analysis and, in particular, the results of quantitative national surveys on social responses to digital media as a source of information and consolidation in the context of the COVID-19 pandemic crisis. The set of materials and data that are analyzed refer to the period of lockdown from March 2020 to December 2020. Some of the problems that come in the limelight in the attempts to manage and deal with the pandemic, related to the reliability of information, trust in the institutions and units managing the crisis, the transformation of the pandemic into a factor consolidating society, the quality of digital education, and challenges for digital citizenship.

The results of a national survey conducted in the period of 29 April–03 May 2020 with the method of individual direct online inquiry with 609 respondents over the age of 18 from all over the country are presented. The study was conducted by a team of the Department of Sociology SWU "N. Rilski" led by Prof. Dobrinka Peycheva. The other quantitative national survey we are considering was conducted from May–June 2020 by the Institute for Educational Research, it is focused on the effectiveness of digital learning in the COVID-19 situation. The survey was conducted at the end of the 2019/2020 school year and covers 135 schools across the country. It involves 4448 students from 5th to 12th grade, 5403 parents, 1885 teachers, and 135 principals. The sample reflects the structure of the school system in Bulgaria.

The methodology used is aimed at showing that: (1) Training is a prerequisite for building digital skills and literacy, for the promotion of digital citizenship; (2) in terms of social crises-such as COVID crisis digital skills and media are an important source of information and connectivity among people, thereby contributing to a sense of perspective and balance.

## 4. Results

### 4.1. Focus Groups

Most of the respondents in three focus groups in 2019 (before COVID-19) have been able to define and evaluate digital literacy, which is identified with gaining experience of a skill or a group of skills, that most often include the ability to use computers and smartphones; the ability to work on various programs in Windows, information search on the Internet, Facebook, Viber, and Skype. The interesting thing about the respondents from the focus group conducted with the pensioners is that almost all of them regularly use Skype or Viber. The reason for the frequent use of these applications is the opportunity for the respondents to communicate with their relatives who are abroad or to get access to any type of information. Some of the respondents have gained experience in digital skills while working, which includes various professional programs such as the design program for CAD architects and engineers.

In all three focus groups, a difference was revealed between different generations in terms of their digital skills formation: "… much of the older generation was enforced to learn how to use the new digital technologies as it was necessary to keep our jobs or to communicate with our children studying or working abroad." One of the major differences between the generations that have been indicated was the frequency of which the digital technology was used, defined for the younger generations as a "routine" or part of their life. The Bulgarian Muslims also agreed with the statement that the main difference between the generations is related to the faster and easier acquisition of skills while working with digital technologies. They summarized that "the young man's world would be unthinkable without the Internet or a smartphone."

It can be said that in today's Bulgarian society, as a result of the use of digital technologies, a crisis in the family is increasing. Family life is becoming more dependent, and communications are becoming more electronically dependent (via Skype, Facebook, Viber, etc.), and the conflicts that have arisen are more symbolic and virtual than physical. The Bulgarian family seems to be losing control of its children in the battle with digital technologies. A solution to this problem could be to direct efforts in promoting the opportunities of religious education in dealing with problematic, critical, and conflict situations, especially for at-risk groups of children and families.

Almost all participants in the three focus groups agreed that digital skills help people receive better information. The internet and the Google application were indicated as an indisputable source of information, where "the answers to every question can be found".

There was a dispute and interesting contrast in opinion among the participants in the focus group conducted with the young mothers. There are two main cores around which the views were united, excluding the respondents with no opinion. Some of them believe that digital skills do not really foster the development of civil society. As an example was given the failure of organizing protests on Facebook against the Government. Other respondents supported the opinion that Facebook enables large groups of people to fight for a specific cause.

The great interest of the interviewees and the many different opinions in the three focus groups provoked the question of the possible "menace" of digitization in society. The young mothers shared the view that the frequent use of computers and new technologies, could be a real "menace" because "the need for human intervention in some areas is reduced and unemployment is increased accordingly". Therefore, digitization in the long run will require the disappearance of some professions. Another problem with the digitization of society was the problem of communication between people and especially physical communication. Parallel worlds have been created in the internet space that are replacing reality, the relationships in the family are lost, and the relationships between parents and children are broken.

The pensioners agreed that there was a real "menace" caused by digitization in society as it would replace the leading up to now human factor as a result of "machines and computers use". In this connection is the topic of vocational training, see Jeleva, Nakova, 2019 [21]. The Bulgarian Muslims also believe that digitization can be a "menace" to society. Misunderstanding and lack of information about digital technologies is a real "menace". Like any technology, deliberate abuse is possible in the digital society, which can be a result of overtrusting people. In all three focus groups, there were respondents who said that digitization cannot cause a "menace" to society, "on the contrary, it is a natural process of human development and the benefits should be sought, not the threats".

The main highlights of the second type group discussions were related to the lockdown period, where the importance of digitization digital citizenship is especially highlighted by the students. They said that digital skills help people to be informed:

"They definitely help because they give access to a very large database of information that can be used for different things" (female, 21 years old).

Digital skills contribute to career development:

"Definitely yes, even this is a requirement for almost all better business positions" (female, 21 years old).

Digital skills support the development of civic consciousness:

"Yes, for example, irregularities are filmed, posted, for example, on social networks or blogs... and this somehow influences people's social behavior. I feel responsible to take pictures of things that are wrong and to share them with others so that there is a reaction to them" (male, 20 years old.).

"I myself am interested in the environmental topic, and receive information on social networks about various events, we have a group in which such information is shared." (Female, 19 years old).

Thus, digital skills and access to information become a social regulator.

COVID's situation has created a serious challenge—the need for secondary and higher education to move to an online environment. In these conditions, mobile learning in its various forms, and variants have become a reality in the last 2020 year. During the COVID-19 pandemic Bulgarian educational system meets the challenges posed by the need to restructure training and universal penetration of e-learning. The access to online training was the most discussed issue and is still so throughout the period of restrictions.

The topics discussed in the focus group were related to:

- The effectiveness of ongoing digital learning;
- the readiness of the educational system as a capacity and resources for online learning;
- the challenges of online learning.

According to the students, learning from a distance and held in an electronic environment is effective in nature, because the lectures through the different platforms are conducted as fully as in a face-to-face environment.

"The lectures follow the set time as a start and end time, the teachers enter the platforms and present the topics they have provided, then there is time to ask questions from us and ask if something is not clear" (male, 19 years old).

"For me there is no difference in what is taught, the things are the same, whether it will be in the university hall or at home in front of a laptop, there is no difference" (woman, 21 years old).

"Online learning is more effective because teachers give more materials, links to publications than in face-to-face form. Besides, you don't waste time traveling to the university, everything is electronic and so one mobilizes more" (male, 21).

It was shared that digital learning reproduces what is in terms of ambition, motivation, and activity, i.e., students who have studied and taken learning seriously, regardless of the conditions and environment, reproduce and retain an identical predisposition. This means that learning is a matter of personal decision, activity, and motivation; the environment itself, traditional or high-tech, does not have such a big impact. However, the participants in the focus group definitely evaluate online learning as a digital transformation in education, because until now, even if there were specific examples of conducting this type of training, they were isolated cases. The whole educational environment in terms of technology had to be restructured, new rules of organization, participation had to be introduced in a short time, platforms had to be imposed, and the teachers themselves had to be qualified and reach the necessary level for participation and management of the technological environment. According to the respondents:

"not all teachers are equally good and do equally well with the management of the technological system" (male, 20 years old).

"it can definitely be said that young teachers are more adequate and it is easier for them to operate with digital platforms and their rules and their implementation" (female, 21 years old).

The main challenges related to digital learning concerned the provision of resources, access to high-speed Internet, and the availability of good mobile devices through which to download and upload any information from various sites, sources, and platforms. Another challenge is the need for social contacts. Online learning creates a certainty in the social sense. Young people need to meet their fellow students, their peers, to maintain an open life in which meetings and communication take place in a living environment. From this point of view, most of the participants in the group discussion emphasized that the hybrid form of training, which provides classes that, on the one hand, take place online and also have face-to-face classes, is a good opportunity to balance in creating an appropriate discussion environment and interactive learning. In this sense, digital learning has made a serious breakthrough in the education system and created the preconditions for a new modern attitude to learning. Respondents stressed that digital skills are a necessary prerequisite not only for finding a job, but digitization contributes to people's social and civic consciousness. Increasingly, various social events go through the Internet and chats, which are used as a space to share opinions, but also to activate people's behavioral empathy.

Young people say that due to digitalization received a big advantage associated with promoting the causes and take practical actions, which are an important part of their civic consciousness. Thus, the idea of social responsibility is refracted through digital citizenship.

### *4.2. Analysis of Surveys*

### 4.2.1. Digital Education at the Secondary School in the Conditions of COVID

As we have already emphasized, the main prerequisite/precondition for the formation of digital citizenship is the development of digital literacy and digital skills. In the situation of COVID-19, Bulgarian education became digital. In contrast with the universities, where the digitalization of education was relatively more successful, in high school this was not been the case.

The other quantitative national survey we are considering was conducted from May–June 2020 by the Institute for Educational Research, and it is focused on the effectiveness of digital learning in the COVID situation. The survey was conducted at the end of the 2019/2020 school year and covers 135 schools across the country. It involves 4448 students from 5th to 12th grade, 5403 parents, 1885 teachers, and 135 principals. The sample reflects the structure of the school system in Bulgaria.

One of the main topics of this survey was access to equipment. The availability of electronic devices is essential for the effectiveness of distance learning because it provides an opportunity for interaction between teachers and students, and between students themselves. Individual access to equipment was measured through questionnaires for students and teachers. In 86% of schools, all teachers have a computer, laptop, or tablet in their home. In the remaining 14% of schools, between 1% and 16% of teachers do not have access to an electronic device at home. 68% of the schools have taken measures for additional resource provision of the teaching process by providing equipment (electronic devices) for personal use by the teachers (Figure 1). While more than 80% of urban schools have provided teachers with additional equipment, only 49% of rural schools have provided their teachers with electronic devices for personal use. At the same time, the data show that the share of teachers in villages who do not have their own computer (or other electronic device) is higher than that of teachers without devices in cities. To the extent that the teaching process (and its resource provision with equipment in the conditions of distance learning) can be considered as a direct determinant of learning, the relatively weak resource provision of the teaching process in villages can be considered as a possible difficulty in achieving of the effectiveness of distance learning in an electronic environment in these settlements.

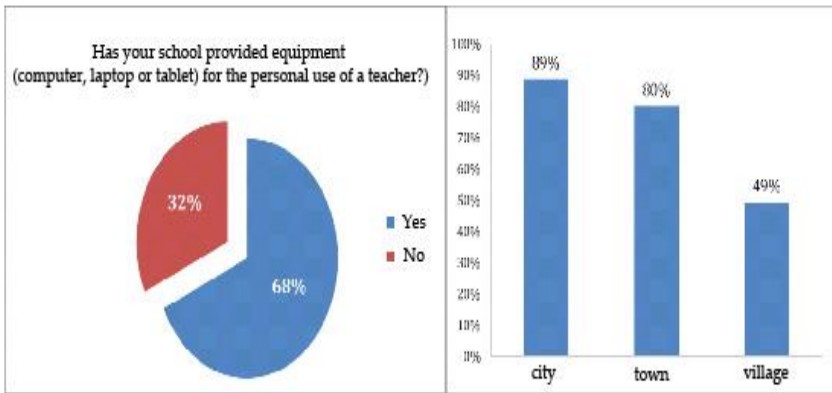

**Figure 1.** Share of schools that have provided equipment (electronic devices) for personal use of teachers.

Providing conditions for students to participate in distance learning in an e-learning environment by supporting those who do not have access to electronic devices at home is a key factor in supporting learning and reducing the manifestations of educational inequalities. Half of the schools have made a targeted effort to provide electronic devices to students in need, thus ensuring the necessary minimum conditions for the integration of technologies to support their learning (Figure 2). 60% of schools in small towns and 49% of schools in villages have taken measures to ensure the learning of their students by providing them with electronic devices (mainly tablets and laptops).

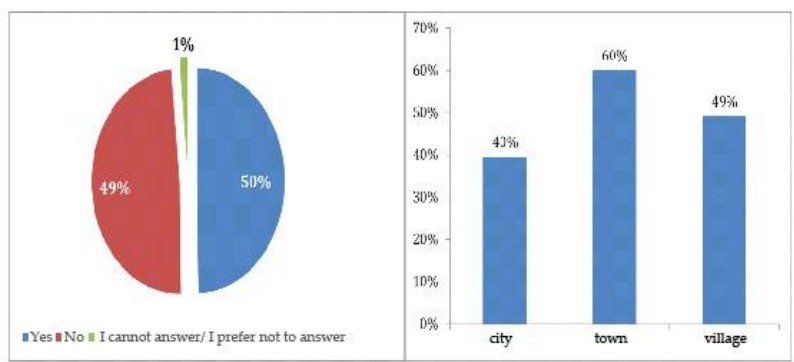

**Figure 2.** Share of schools that have provided electronic devices for personal use to their students.

Internet access and the availability of a quality internet connection are key conditions for conducting distance learning in an electronic environment, for the application of a synchronous process of teaching and learning, and for maintaining active relationships between teachers and students and between students themselves. Providing the opportunity for (a) interactive communication between students and teachers (including teaching, discussion of the material, learning support, asking and answering questions, encouraging students, providing feedback, etc.) and (b) student cooperation in the learning process are considered key conditions for effective distance learning [22].

The data show that in 83% of schools all teachers have permanent access to home internet. However, it should be noted that in 3% of schools, there is not a single teacher who has internet at home, and $\frac{3}{4}$ of these schools are located in the villages (Figure 3). In practice, in almost 6% of schools in the villages, no teacher has access to home internet. In these schools, there are no prerequisites for teaching in an electronic environment from the teacher's home. This implies limited opportunities for interaction between students and teachers, and creates preconditions for (1) skipping material and lagging behind in learning, (2) accumulation of inefficiencies in the learning process (especially if parents do not have the opportunity to support their child in learning), and (3) increasing educational

inequalities. Teachers in most schools have used more than one distance learning platform in an e-learning environment. The most widely used platforms are MS Teams, Google Classroom, School, Zoom, and Moodle. In many schools, the platforms were changed in the process of work. About 8% of schools used only applications (Viber, Messenger) and social networks (Facebook, Instagram). Different schools have used different approaches for choosing platforms. Despite the considerable freedom of choice granted to them, for 48% of the schools, the leading factor in the choice of platforms was the recommendations of the Ministry of Education and Science. Less than half (46%) of schools have taken into account the preferences and capabilities of students, although this should be a leading consideration given the fact that students' perceptions of the usefulness of a platform and their ability to work with it are key conditions for students' engagement and active participation in online learning [22]. The access and use of various educational resources, as well as the creation and use of appropriate didactic materials by teachers, are essential for the effectiveness of distance learning and learning in an electronic environment.

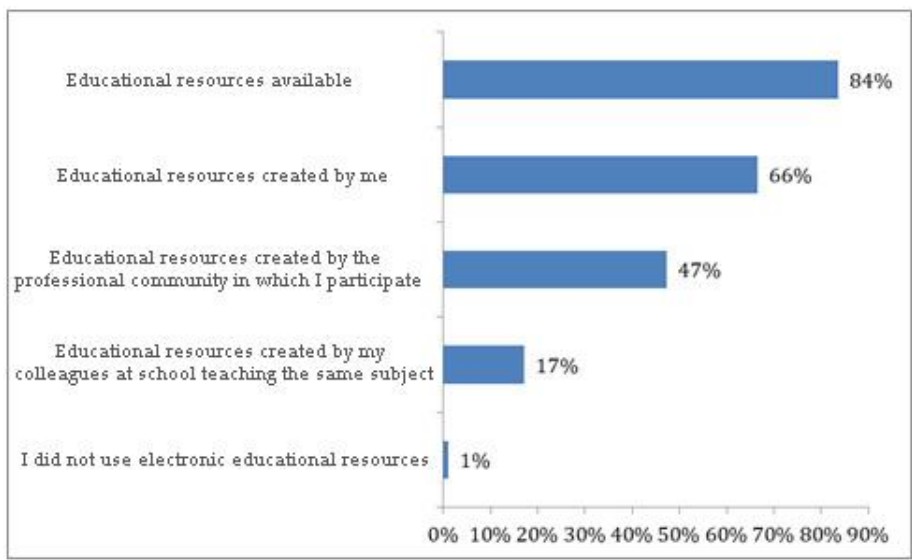

**Figure 3.** Source of the used electronic educational resources (share of teachers who have chosen the respective opportunities).

The data shows that teachers have had the opportunity to use a variety of resources from different sources, as well as to actively create them. The vast majority (84%) of teachers used available online educational resources, and 66% created their own resources (Figure 3). Of the available educational resources, teachers most often used presentations (83% of teachers), video lessons, and other video resources (82%). 72% of the teachers used ready-made didactic materials (Figure 3). Among the "other" educational resources used by teachers are online educational games and quizzes, interactive real-time tasks and exercises, online tests, ready-made worksheets, photos, graphs and tables, and more. Some teachers have also actively used international material-sharing sites. The participation of teachers in professional teaching communities has also made a significant contribution to the provision of the learning process with e-learning resources—for almost half of the teachers, this has provided access to educational resources created within the professional community. At the same time, the sharing of resources between teachers from the same school teaching the same subject seems to be a less common practice—only 17% of teachers used educational resources created by their colleagues in the school teaching the same subject (Figure 3).

### 4.2.2. Quantitative National Survey of Digitization in Crises

One of the main things in digital citizenship is the ability to sift through reliable information. The COVID-19 situation was an example of the huge growth/increasing of information in the digital media about the pandemic, its origin, treatment, spread, number of infected. There was a lot of fake news, which made it difficult for people and created great chaos, uncertainty, and ambiguity about what was really happening. The answers that stood out in the online survey conducted in April–May 2020 are indicative. About 80% of respondents believe that they do not have the skills to recognize false information (Figure 4).

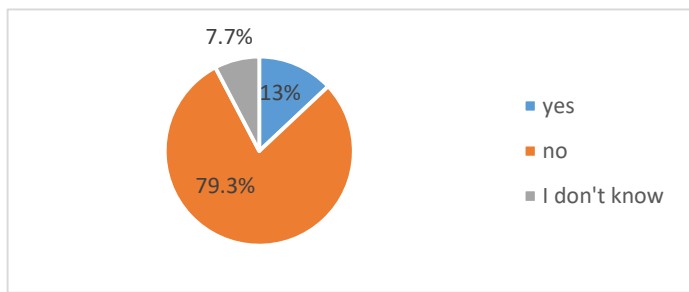

**Figure 4.** People can successfully sift false from true information provided by the media, including social networks.

The emergence and dissemination of specialized digital tools and videos to deal with in recent years, as well as the European project "Media Literacy for All", are an expression of Europe's concern about the spreading "large-scale misinformation of citizens, including through misleading or downright false information" [4]. The confrontation of society as a whole with the problem of the reliability of the information presented in the media is an indicator of the degree of the ecological nature of our social system, respectively, of the morality of the institutions in particular, as well as the declining trust in them.

In addition, taking into account the lack of ability to screen true information as an indication of the need for new digital literacy with its inherent critical dimensions. Websites of both international health organizations and world news agencies, as well as their national counterparts, including blogs of many public figures and institutions, have become popular. The abundance of materials—from less to very sensational, from more specialized to widely popular, from authorial to pseudo-authorial, from fully supporting the actions of the headquarters and the government to denying them—accompany the media daily. Television has proved to be a strongly predominant main source of information for people and an on-screen means of recreating images and pictures of existing pandemic realities, regardless of the source (Figure 5).

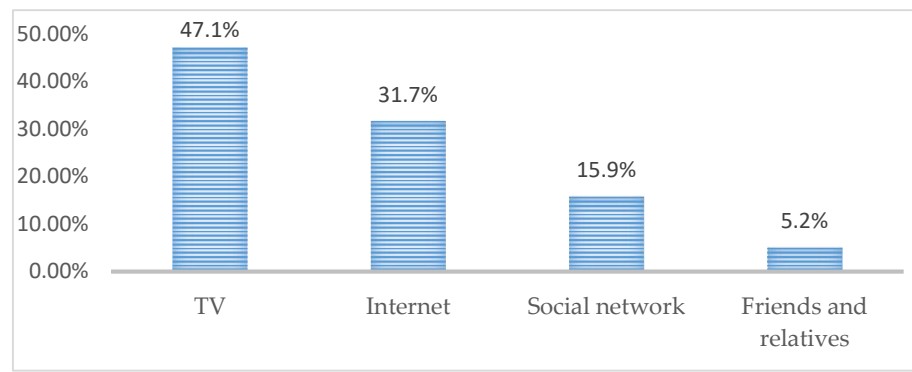

**Figure 5.** Where are you basically informed about COVID-19 from?

It is important to note the high share of Internet sites and social networks as a source of information about COVID-19.

All respondents are characterized by the search for and use of sources that complement the "picture" of the pandemic. 62.6% say that they read materials from various media sources, including specialized publications providing information about COVID-19. Additional materials are sought in social networks and about 20% of respondents say they discuss the topic COVID-19 in social networks. In general, websites, blogs, and social media have an informational presence in the media space and are accepted as a source of information.

Social media (Facebook, Instagram, Twitter, TikTok, Snapchat, etc.) also correspond to information needs. In answering the question: What needs are met most strongly in social networks—informational are leading, indicated by 74.4% of individuals. After the information are the needs for maintaining social contacts and friendships, for overcoming boredom, for causes, etc. The study shows an internal restructuring of needs in social networks that are not typical for other periods.

However, communication through social networks is not considered as a compensator for physical contacts. The majority of respondents share that only partially compensates for their need for live contacts through social networks, which are assessed as a social vent, a satisfactory environment for online social contacts, and more. The prevailing view is that social networks cannot compensate for physical isolation (Figure 6).

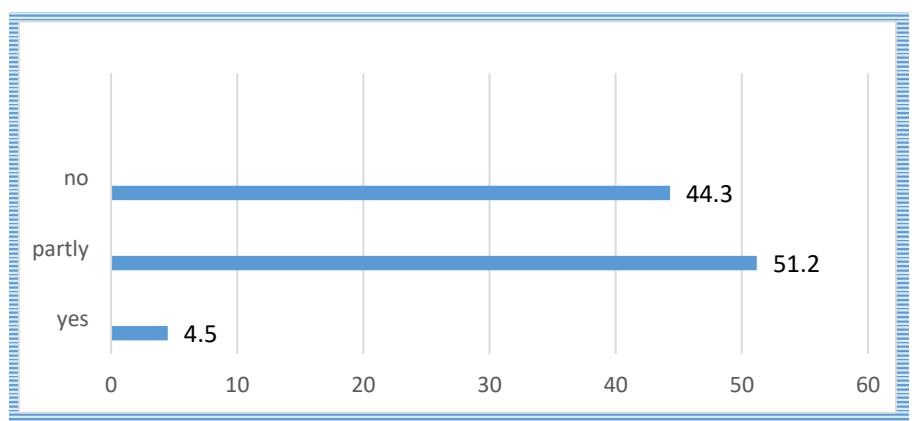

**Figure 6.** Do social networks compensate for physical isolation for communication with the desired people?

## 5. Discussion

The vision of education to be a foundation for building digital skills and literacy, for promoting digital citizenship, is emerging. Acquired competencies are associated with the formation of sensitivity, behavior, and culture, but also with commitment to the environment in the broadest sense by achieving understanding and activity, creative attitude and awareness of the need to transform existing models. Among the factors that could turn into potential barriers and obstacles to the success of the training for digital skills are: Lack of experience and resources, lack of training opportunities—further training for teachers. It highlights the opportunity for constructive involvement of the non-private sector as a partner in the process of digital citizenship. Stimulating creativity and critical thinking, promoting independence, combining social skills, cognitive abilities, and overall emotional and intellectual competence are trends that can be seen as a possible resource for the development of digital citizenship. Another aspect of the discussion are young people who are undoubtedly perceived as valuable capital in the formation of digital skills and citizenship. The change in the culture of understanding and attitudes of young people towards the digital environment means a change in values and lifestyle. This process that affects the nature of interactions between individuals in different social communities, in

society as a whole, and is associated with the development of social capital, and with the development of competencies for critical thinking and self-reflection.

## 6. Conclusions

This article was focused on showing the essence of digital citizenship as one of the underlying theoretical and practical structures of global society. In conditions of total digitalization, digital citizenship becomes the key to individual behaviors, because what integrates them are the collectively shared values of civil society. Thus, it provides public legitimacy of digital citizenship, that creates expectations of reciprocity and mutual obligations. In fact, digital citizenship as a concept in the scientific literature means justifying the civic people that becomes an incentive to contextualize their requirements and expectations of the political environment. In this aspect, this article has a research contribution related to the substantiation of digital citizenship as a theoretical and applied side of the global society, as well as to the enrichment of the analyzes and conclusions about its essence. On the other hand, the article has several practical achievements related to the display of applied nature of digital skills and citizenship. The main prerequisite for digital citizenship is the digital skills of individuals. One of the conclusions of the article is that digital literacy during social crises is crucial for maintaining the normal course of ongoing events and their interpretations. In this context, digital citizenship contributes to the control of community understanding and individual practices in the global pandemic trajectory. It was in this direction that the different results of qualitative and quantitative research presented in the article showed that people in different contexts appreciate the need for digital skills that need to be improved in parallel with improving the technological characteristics of different digital devices. In this context, digital citizenship contributes to the control of community understanding and individual practices in the global pandemic trajectory. The results of the study related to the COVID situation and the conclusion showed that digital media and social networks are the main source of information. Another important topic to emphasize the applicability of digital skills was the online training that took place after the blockade with COVID-19. It is assessed as a digital transformation in education because it has imposed changes in the technological plan in the overall educational environment, structures, and organization of teaching, and in general transforms the nature of learning itself. Mobile learning requires technological support, availability of high-speed Internet, and mastering the rules for working with platforms in a digital environment. An important conclusion is that gradually all participants in education—students and teachers have adapted to online learning and its requirements, but in general the preference is to supplement it with face-to-face training that provides live contact and interaction. This article has shown that digital citizenship means a change in the culture that every organization must make in order to survive. This is the current process of creating activities taking place in a changing digital environment in the conditions of rapid transformations. The digital world is changing rapidly and constantly and requires learning new things, which is why not only the skills of different social groups must be assessed, but also the quality of education that digital knowledge must provide them, as well as access to resources, to the Internet and technology. In this context, it is necessary to develop research on digitalization and its consequences in the social environment, related to the expectations, attitudes, skills, and assessments of modern people.

**Author Contributions:** Conceptualization, V.M. and V.L.; methodology, V.M. and V.L.; formal analysis, V.M. and V.L.; data curation, V.M. and V.L.; writing—original draft preparation, V.M. and V.L.; writing—review and editing, V.M. and V.L.; visualization, V.M. and V.L.; supervision, V.M. All authors have read and agreed to the published version of the manuscript.

**Funding:** This research was funded by the project "Digital Media Literacy in the Context of a Knowledge Society: Status and Challenges" КП-06-Н25/4, funded by National Science Fund. This publication is based upon work from the project "Digital Media Literacy in the Context of a Knowledge Society: Status and Challenges".

**Institutional Review Board Statement:** The study was conducted according to the guidelines of the Declaration of Helsinki and approved by the Institutional Review Board (or Ethics Committee of the University of Girona) under the code: CEBRU0001-2018 (6 April 2018).

**Informed Consent Statement:** Informed consents were gathered from all subjects involved in the study.

**Data Availability Statement:** The data shared are in accordance with consent provided by participants on the use of confidential data.

**Acknowledgments:** This article is based on the project "Digital Media Literacy in the Context of a Knowledge Society: Status and Challenges" КП-06-Н25/4, funded by National Science Fund.

**Conflicts of Interest:** The authors declare no conflict of interest.

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
