# Peer review of "Digital Citizenship and Digital Literacy in the Conditions of Social Crisis"

_computers, doi:10.3390/computers10040040_

Round 1

Reviewer 1 Report

I am pleased to have the opportunity to review this research paper. This study attempted to explore digital citizenship and digital literacy in the Conditions of Social Crisis. Although the topic of this research study is interesting and fits within the journal scope, I think authors should apply the comments indicated below to increase the quality of research justification, contributions, and findings.

The paper research gap is very general and lacked alignment to the research findings, no discussion was provided to derive the implication from. Theoretical and pragmatic implications are vague and need to be better aligned with this paper's theoretical underpinnings and proposed process. Furthermore, there is insufficient support and weak arguments in support of the objective that is proposed as well as the model developed. In the final part of the introduction, the manuscript structure should be summarised as well as the objectives proposed, originality, and gap that would be covered. Use this citation to improve the introduction section a cite it:

Saura, J.R. (2020). Using Data Sciences in Digital Marketing: Framework, Methods, and Performance Metrics, Journal of Innovation and Knowledge, 1(2020). doi: https://doi.org/10.1016/j.jik.2020.08.001

Please consider this structure for the manuscript final part.

Conclusion

Managerial Implication

Practical/Social Implications

Limitations and future research

The conclusion section should emphasize and explicitly spell out the contribution to the theory that the analysis brings about. Practical implications can be summarised in one paragraph. The same for the limitations and further research.

There is no discussion section. This needs to be a coherent and cohesive set of arguments that take us beyond this study in particular, and help us see the relevance of what the authors have proposed.  The authors need to contextualize the findings in the literature and need to be explicit about the added value of your study towards that literature.

Also, other studies should be cited to increase the theoretical background. Findings should be contextualized in the literature and should be explicit about the added value of the study towards the literature. Please check this manuscript and make a citation to them. This is the weakest part of the article.

Saura, J. R., Ribeiro-Soriano, D., & Palacios-Marqués, D. (2021). From user-generated data to data-driven innovation: A research agenda to understand user privacy in digital markets. International Journal of Information Management, 102331. doi: 10.1016/j.ijinfomgt.2021.102331

Mossberger, K., Tolbert, C. J., & McNeal, R. S. (2007). Digital citizenship: The Internet, society, and participation. MIt Press.

Hollandsworth, R., Dowdy, L., & Donovan, J. (2011). Digital citizenship in K-12: It takes a village. TechTrends55(4), 37-47.

Hintz, A., Dencik, L., & Wahl-Jorgensen, K. (2017). Digital citizenship and surveillance| digital citizenship and surveillance society—introduction. International Journal of Communication11, 9.

Saura, J. R., Ribeiro-Soriano, D., & Palacios-Marqués, D. (2021). From user-generated data to data-driven innovation: A research agenda to understand user privacy in digital markets. International Journal of Information Management, 102331. doi: 10.1016/j.ijinfomgt.2021.102331

Questions to be answered in conclusion:

What practical/professional and academic consequences will this study have for the future of scientific literature (theoretical contributions)?

Why is this study necessary? Again, the authors should make clear arguments to explain what is the originality and value of the proposed model. This should be stated in the final paragraphs of the introduction and conclusion sections.

Please re-design Figure 1 and Figure 2.

Author Response

We have filled out the Introduction by pointing out what the originality and significance of the article is: lines: 45-48; 53-59

We have revised and filled out the Digital Citizenship section, including recommended by the reviewer 1 references; lines: 78-81; 101-107.

We have added a new paragraph “Discussion”: lines: 533-550

We have rewritten the Conclusion, emphasizing the guidelines given by reviewer 1 - to show the contribution of the article in theoretical and practical terms, and the future author's guidelines.: lines: 553-593.

We have filled out the cited literature and references.

We have made a complete revision of the article in terms of achieving greater coherence and connection between the paragraphs themselves and the overall content focus; by shortening some paragraphs: lines: 252-254; 257-261, 594-636 and we have added new ones: lines: 238-242; 377-381.

We have improved the quality of the figures.

Reviewer 2 Report

The abstract presents well the context of the paper.

The paper is overall good and interesting in the context of the covid pandemic.

The paragraph on basic considerations should be renamed with regard to the content.

The discussion should have a proper paragraph.

It is also unsufficiently clear the role of the national survey connected to the focus groups.

Author Response

We have added a new paragraph “Discussion”: lines: 533-550

We have rewritten the Conclusion, emphasizing the guidelines given by reviewer 2 - to show the contribution of the article in theoretical and practical terms, and the future author's guidelines.: lines: 553-593. We have made a complete revision of the article in terms of achieving greater coherence and connection between the paragraphs themselves and the overall content focus; by shortening some paragraphs: lines: 252-254; 257-261, 594-636 and we have added new ones: lines: 238-242; 377-381.

Round 2

Reviewer 1 Report

The authors have addressed all my comments correctly. This reviewer has no additional comment.